# Chromosome Translocations as a Driver of Diversification in Mole Voles *Ellobius* (Rodentia, Mammalia)

**DOI:** 10.3390/ijms20184466

**Published:** 2019-09-10

**Authors:** Svetlana A. Romanenko, Elena A. Lyapunova, Abdusattor S. Saidov, Patricia C.M. O’Brien, Natalia A. Serdyukova, Malcolm A. Ferguson-Smith, Alexander S. Graphodatsky, Irina Bakloushinskaya

**Affiliations:** 1Institute of Molecular and Cellular Biology, Siberian Branch, Russian Academy of Sciences, 630090 Novosibirsk, Russia; 2Department of Natural Sciences, Novosibirsk State University, 630090 Novosibirsk, Russia; 3Koltzov Institute of Developmental Biology, Russian Academy of Sciences, 119334 Moscow, Russia; 4Institute of Zoology and Parasitology, Academy of Sciences of Tajikistan, Dushanbe 734025, Tajikistan; 5Cambridge Resource Centre for Comparative Genomics, Department of Veterinary Medicine, University of Cambridge, Cambridge, CB3 0ES, UK

**Keywords:** chromosome painting, karyotype, mole vole, speciation, subterranean rodents

## Abstract

The involvement of chromosome changes in the initial steps of speciation is controversial. Here we examine diversification trends within the mole voles *Ellobius*, a group of subterranean rodents. The first description of their chromosome variability was published almost 40 years ago. Studying the G-band structure of chromosomes in numerous individuals revealed subsequent homologous, step-by-step, Robertsonian translocations, which changed diploid numbers from 54 to 30. Here we used a molecular cytogenetic strategy which demonstrates that chromosomal translocations are not always homologous; consequently, karyotypes with the same diploid number can carry different combinations of metacentrics. We further showed that at least three chromosomal forms with 2n = 34 and distinct metacentrics inhabit the Pamir-Alay mountains. Each of these forms independently hybridized with *E. tancrei*, 2n = 54, forming separate hybrid zones. The chromosomal variations correlate slightly with geographic barriers. Additionally, we confirmed that the emergence of partial or monobrachial homology appeared to be a strong barrier for hybridization in nature, in contradistinction to experiments which we reported earlier. We discuss the possibility of whole arm reciprocal translocations for mole voles. Our findings suggest that chromosomal translocations lead to diversification and speciation.

## 1. Introduction

The role of chromosome changes in speciation has been discussed for many years [1,2,3,4,5,6]. Chromosome rearrangements disrupt the spatial positioning of chromosomes in the nuclei, and Robertsonian translocations can change intranuclear organization dramatically and alter interactions among chromosomal domains [7,8,9]. In morphologically indistinguishable species, the so-called cryptic species, morphological similarity exists alongside significant differences in genetic markers, such as chromosome structures or specific genes. There seem to be no radical changes in the development of morphological traits, but reproductive isolation appears, which makes such species a good prospective model for studying evolution [10,11,12,13].

Rodents of genus *Ellobius*, or mole voles, are very similar morphologically, due to their adaptation to a specific subterranean style of life. Also, the genus is an excellent example of diversification of sex chromosomes: XX♀-XY♂ in *E. fuscocapillus*, X0♀-X0♂ in *E. lutescens*, and XX♀-XX♂ in *E. talpinus*, *E. tancrei*, and *E. alaicus* [14,15,16,17]. Even before studying chromosomes, some paleontologists considered *E. tancrei* and *E. talpinus* to be distinct species, whose ancestral lines diverged in the Pleistocene [18]. Then, although the species status was confirmed by cytogenetic data [19], until the 1980s *E. tancrei* was referred to as conspecific with *E. talpinus* [20,21]. *E. talpinus* and *E. tancrei* share the same diploid number, 2n = 54, but differ by the fundamental number, NF = 54 and 56, respectively. The centromere reposition which changed the fundamental number probably led to their reproductive isolation [22]. Till now, no chromosomal variability has been shown for *E. talpinus* (2n = NF = 54) [23]. However, *E. tancrei* has one of the most variable karyotypes among mammalian species with diploid numbers varying from 54 to 30 [16,20,24,25,26]. Using data from comparative chromosomal painting, the ancestral acrocentrical karyotype of *Ellobius* was reconstructed [27]. White [2] hypothesized that the evolution of karyotypes in mammals was directed towards a reduction of chromosome number, which Robertsonian translocations can affect.

Three hypotheses were outlined for an explanation of the wide chromosome variability in *E. tancrei* [20]: (i) a low-chromosomal form originated from the ancestral 2n = 54 karyotype by chain mutation processes and intermediate forms kept in distinct populations; (ii) a single sudden emergence of a low-chromosomal form was followed by its hybridization with the original form and subsequent stabilization of chromosomally balanced forms; (iii) multiple and independent translocations of acrocentrics in different populations occurred, so metacentrics can be homologous or non-homologous, unlike in first two cases where the Robertsonian metacentrics must be homologous.

This study aims to describe the karyotype structure of all currently known chromosomal forms of *E. tancrei* from the Pamir-Alay, the zone of greatest chromosomal variation, using chromosome painting and G-banding comparison to determine their paths of evolution.

## 2. Results

The karyotype structure was determined for 379 mole voles from 80 localities across the Pamir-Alay mountains, mostly the Surkhob and Vakhsh rivers valleys (Figure 1 and Figure 2). Among the samples, we found 29 animals from 8 localities with the typical for *E. tancrei* 2n = 54. We compared chromosome painting data for nine specimens (collection numbers 24914, 24915, 24919, 24920, 25613, 25614, 25617, 25620, 25621, see Figure 3 for selected examples) with previously published karyotypes [26,28,29]. Samples demonstrated distinct Robertsonian translocations (Rbs), which could be distinguished by G-banding or chromosome painting. We revealed the existence of three allopatric forms with basic 2n = 34 and specific Rbs. Moreover, scanning G-banded karyotypes of mole voles from the Surkhob and Vakhsh rivers valleys, we revealed numerous hybrids of each of these forms with the original 54-chromosomal *E. tancrei*.

### 2.1. Chromosome Form I

Chromosome form I is located mostly at the northern bank of the Surkhob River from the left bank of the Sorbog River (approximately 70°11′E longitude) at the West to 71°21′ at the East. Unexpectedly, this form crossed the rough wide mountain river and settled at the southern bank also (Figure 1). We collected data on karyotype structure for 160 animals from 31 localities (Figure 2a). Diploid numbers varied from 2n = 53 to 2n = 30 due to the presence of 1 to 24 Rbs in different combinations alongside with stable NF = 56: Rb(1.8), Rb(2.11), Rb(3.18), Rb(4.14), Rb(5.9), Rb(10.19), Rb(6.12), Rb(13.23), Rb(15.22), Rb(16.20), Rb(17.25), and Rb(24.26). All Rb translocations detected in this form by chromosome painting were homologous to Rbs in mole voles with 2n = 30 [26].

On the northern bank of the Surkob River, where mole voles are numerous, we found animals with 2n = 30, 32, 34, and few individuals with 2n = 33 (Figure 1 and Figure 4a,b).

*E. tancrei* with 2n = 54 populate large areas to the North, such as the Fergana and Varzob valleys and migration from these regions to the Surkhob River Valley was possible in the past. Hybridization with the 34-chromosomal form led to the origin of the hybrid zone, and a homozygous form with 2n = 48, 2Rb(2.11), 2Rb(3.18), 2Rb(5.9) (Figure 5a), and more Rbs in the heterozygous state (Figure 5b). There was also an area that was inhabited by mole voles with 2n = 37-48 (Figure 1 and Figure 2). Karyotypes of all these hybrids bore 2Rb(2.11), 2Rb(3.18), 2Rb(5.9), and, in fewer specimens, other Rbs, often in a heterozygous state were found: Rb(1.8), Rb(4.14), Rb(10.19), Rb(6.12), Rb(15.22), and Rb(16.20). In this area, the smallest metacentrics Rb(15.22), and Rb(16.20) were observed in karyotypes with 2n = 40 or lower.

Mole voles with 2n = 32 –34, which inhabited the southern bank of the Surkob River (Figure 4c), formed a narrow hybrid zone with *E. tancrei* 2n = 54. We studied 9 animals, which had 2n = 50–53 due to diverse sets of translocations: Rb(1.8), Rb(2.11), Rb(4.14), Rb(6.12), Rb(10.19), Rb(15.22). The large chromosomes numbers at this site probably emerged due to recent hybridization with *E. tancrei* 2n = 54, which inhabited the southern bank of the Surkhob River from approximately 39°8′ N 70°57′ E up to the left bank of the Muksu River, where *E. alaicus* (2n = 48, NF = 56) settled [30]. We were unable to detect a hybrid zone between *E. tancrei* and *E. alaicus* there.

### 2.2. Chromosome Form II

Chromosome form II with 2n = 34, NF = 56 and specific Rbs were located at the confluence of the Surkhob and the Obikhingou rivers, up to approximately 70°30′E longitude, and in the adjacent mountains to the south, close to the Khozar-Chasma settlement (Figure 1). A total of 96 specimens from 20 localities were karyotyped (Figure 2b). Compared to chromosome form I, only two specific Rbs were revealed by chromosomal painting: Rb(2.18) and Rb(3.11) (Figure 6a). These translocations appeared to be whole arm reciprocal translocations (WARTs) with Rb(2.11) and Rb(3.18), as found in chromosome form I. Other Rbs were identical to metacentrics of the chromosome form I.

We found numerous hybrids closer to the Obikhingou River, their chromosomal numbers varied from 2n = 50 to 2n = 39 due to contacts with the typical *E. tancrei*, 2n = 54, that inhabited mountains near the Chil-Dora settlement. Most of the specimens with 2n = 50 were homozygous for two translocations: Rb(2.18) and Rb(5.9) [25], but some were complex heterozygotes (Figure 6b). The typical for this form translocations Rb(2.18) and Rb(5.9) were detected in all karyotypes studied. There were numerous heterozygotes carrying even up to four translocations (Figure 6c). In hybrids with Rb(2.18) and Rb(5.9) translocations Rb(1.8), Rb(3.11), Rb(6.12), Rb(15.22), and Rb(16.20) were found.

### 2.3. Chromosome Form III

Chromosome form III with 2n = 32–34 was discovered on the right bank of the Vakhsh River, at the area from the right bank of the Sangikar River up 38°47′N 69°51′E, and at the opposite bank of the Vakhsh River, close to the Mienodara settlement (Figure 1). In total, 94 animals from 21 localities were studied (Figure 2c). In karyotypes the animals studied, one Rb was the same as in chromosome form II—Rb(2.18), two others were characteristic for this form only (Figure 7a,b). Chromosome 3 underwent translocation to chromosome 24, forming Rb(3.24), and chromosome 11 translocated to 25, creating Rb(11.25). Other Rbs were shared with I and II chromosome forms.

Unlike other forms, chromosome form III formed a hybrid zone, bordered by rivers. In the confluence of the Sangikar and Sorbog Rivers, 17 specimens from two localities with 2n = 35–41 were studied, with a different combination of all Rbs. High heterozygosity supported the presumption of hybridization between the low-chromosomal form III, 2n = 34 with the original form, 2n = 54 (Figure 7c). Animals were numerous in 1982, but after the earthquakes and powerful mudslides in 2002 along with anthropogenic pressure, the population decreased dramatically.

The right bank of the Vakhsh was inhabited by mole voles with 2n = 32 and some 2n = 33 and 34, which were found from the confluence of the Surkhob and Obikhingou Rivers and over 25 km to the south. On the opposite bank of the river, we discovered hybrids (26 specimens from five sites) with 2n = 33–35, which were located close to man-made bridges.

## 3. Discussion

Our results revealed a high level of heterozygosity in the samples studied. We summarized all data as histograms (Figure 2). Two-colored columns indicated heterozygous specimens. In the II zone we found a single mole vole with four heterozygous Rbs: 20422♂ 2n = 44 1(Rb1.8), 1(Rb4.14), 1(Rb6.12), 1(Rb10.19) (Figure 6c), and three animals which were heterozygous by three Rbs in different combinations: 20423♀ 2n = 43 1(Rb1.8), 1Rb(3.11), 1Rb(4.14); 20425♀ 2n = 43 1(Rb1.8), 1Rb(2.18), and 1(Rb4.14); 21645♂ 2n = 43 1(Rb4.14), 1(Rb6.12), and 1(Rb10.19). The high level of heterozygosity is apparent if histograms for samples from the northern (form I) and southern (form II) banks of the Surkhob River are compared (Figure 2a,b). Both hybrid zones originated as contact zones of low-chromosomal forms with typical *E. tancrei*, 2n = 54. The similarity of diploid numbers masks differences in the karyotype, as in hybrids with 2n = 46 (Figure 5b) or 2n = 50 (Figure 6b).

The three chromosome forms shared partial homology in Rbs: 2.11–2.18–3.18–11.25–17.25. Such monobrachial homology could arise by the independent translocations in different forms or because of specific mutations (whole arm reciprocal translocations, WARTs) lead to reciprocal translocations occurring between two bi-armed non-homologous chromosomes. After the term WART was introduced by Winking [31], this type of mutation was rarely reported as the mechanism is elusive. A single case was described for the spontaneous occurrence of the WART in a wild-derived house mouse [32].

Possible WARTs have been reported in *Sorex araneus* (for review see [33]), different species of *Mus* [34,35,36,37] and some other species. The problem of explaining the occurrence of these translocations has not yet been solved; nevertheless, the presence of monobrachial homology in some cases is easier to explain by WART. In *Mus* and *Sorex* rapid chromosomal evolution is evident due to the existence of numerous chromosomal races. We presumed that the previously described *E. tancrei* with 2n = 48 and two variants of 2n = 50 with monobrachial homology demonstrate an initial stage of speciation due to the meiotic disturbances in hybrids [22,29,38]. In forms with 2n = 34 with numerous Rbs and several non-homological translocations, the meiotic failure is even more predictable. The absence of hybrids even in contact areas indicates the emergence of reproductive barriers for the three chromosomal forms in Pamir-Alay.

We suggest that speciation is currently underway in mole voles. A specific Rb (3.10) was fixed in *E. alaicus* in 30 years [30]. In *E. tancrei* some small chromosomes are not involved in Rb translocations: chromosome 17 in forms II and III and chromosomes 21 and 26 in form I. However, these small chromosomes have a potential for further translocations, as demonstrated for autosomes 24 and 26 in karyotype with 2n = 30 (chromosome form I) [26].

Chromosomal variability of *E. tancrei* is confined to the Pamir-Alay, unlike *Sorex* and, especially, *Mus*, different forms of which are distributed worldwide. The reason given for this by Vorontsov and Lyapunova [39] was explosive chromosomal speciation in seismic active regions. High seismic activity in the Pamir-Alay [40] could lead to sudden rearrangements of the mountain topography; this might isolate small populations with specific mutations and promote diversification [41]. Variability of diploid numbers might have emerged by hybridization inside this restricted mountain area. Distinct hybrid zones, which we described here, may have originated due to contacts of forms with partially homologous metacentrics with the original *E. tancrei*, 2n = 54. Migration for subterranean mole voles is possible in river valleys; the map demonstrates possible geographical connections for forms with 2n = 54 and distinct 2n = 34 (Figure 1). Fast evolutionary changes might be further accelerated or could disappear, because of seismic activity and anthropogenic pressure. In 2008 we were unable to find animals in the confluence of the Sorbog and Sangikar rivers. In 2018, a valley, where the 30-chromosomal form was detected, was built up and plowed and the number of animals decreased dramatically.

Chromosome painting allowed us to reveal the variability and verify the hypothesis for the independent and repeated chromosome translocations for mole voles in the Pamir-Alay. The origin of monobrachial homology is a route to speciation [42]. The emergence of such a homology leads to fertility disorders or sterility of hybrids, which provides isolation of gene pools. Such cases are rare, and *E. tancrei* appears to be one of the most valuable models for studying chromosomal speciation.

## 4. Materials and Methods

Chromosome sets for 379 specimens of *E. tancrei* mole voles from 80 localities from the Pamir-Alay mountains (Figure 1 and Figure 2) were analyzed. Chromosome slides and suspension, prepared from bone marrow (Ford and Hamerton, 1956), were collected in 1981, 1982, 1985, 2008, and 2010 for the cytogenetic collection (a part of the joint collection of wildlife tissues for fundamental, applied, and environmental researches of the Koltzov Institute of Developmental Biology RAS, state registration number AAAA-A16-116120810085-1, Core Centrum of the Koltzov Institute of Developmental Biology RAS, state registration number 6868145). For cross-species chromosome painting (9 specimens from 5 localities) we used cell lines from the IMCB SB RAS cell bank (“The general collection of cell cultures” №0310-2016-0002) established following previously published protocols [43,44] and sets of paints derived from the flow-sorted chromosomes of the field vole *Microtus agrestis* [45]. Cross-species chromosome painting was performed according to protocols [46,47]. G-banding was carried out by trypsin digestion [48]. Karyotypes were checked following the nomenclature of *E. tancrei* chromosomes [22,26].

Images were captured with VideoTesT-FISH 2.0, VideoTesT-Karyo 3.1. (VideoTesT, St. Petersburg, Russia) and Case Data Manager 6.0 (Applied Spectral Imaging Inc., ASI, Carlsbad, CA, USA) software with ProgRes CCD (Jenoptik, Jena, Germany) or ASI CCD camera on an Axioskop 2 plus (Zeiss, Oberkochen, Germany) microscope with filters for DAPI, FITC, and rhodamine. Hybridization signals were allocated to specific chromosome regions defined by G-banding patterns captured with the CCD camera. Giemsa-stained and G-banded metaphase chromosomes were studied with an Axioskop 40 (Zeiss) microscope and captured with a CMOS camera. Images were processed using Paint Shop Photo Pro X3 (Corel, Ottawa, ON, Canada).

All applicable international, national, and/or institutional guidelines for the care and use of animals were followed. All experiments were approved by the Ethics Committee on Animal and Human Research of the Institute of Molecular and Cellular Biology, Siberian Branch of the Russian Academy of Sciences, Russia (order No. 32 of 5 May 2017). This article does not contain any studies with human participants performed by any of the authors.

## Figures and Tables

**Figure 1 ijms-20-04466-f001:**
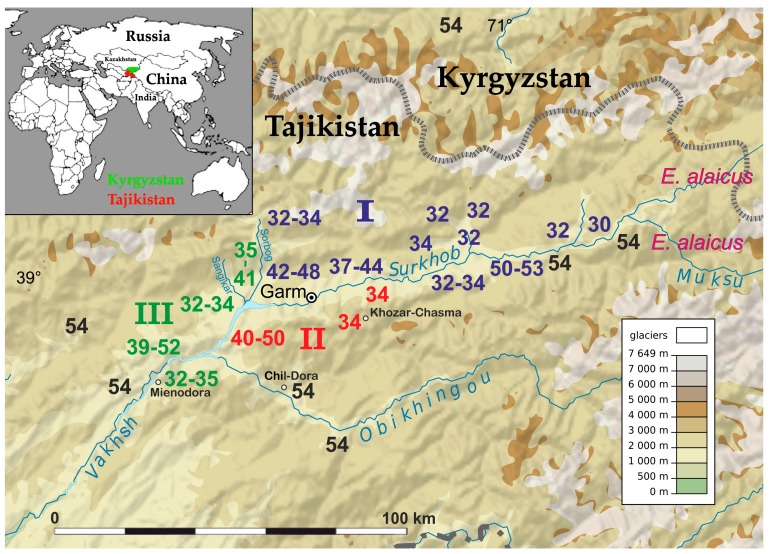
Map of the distribution of the *E. tancrei* chromosome forms found in Pamir-Alay (Tajikistan). Diploid numbers are colored in blue, red, and green for I, II, and III chromosome forms respectively. The original form 2n = 54 is colored in black. The western part of the *E. alaicus* range is marked in pink. The state border of Tajikistan and Kyrgyzstan is marked as a broken line. Geographical connections of different forms are possible through valleys at altitudes below 3500 m above sea level. A fragment of the world map showing the location of the studied sites is placed at the top left.

**Figure 2 ijms-20-04466-f002:**
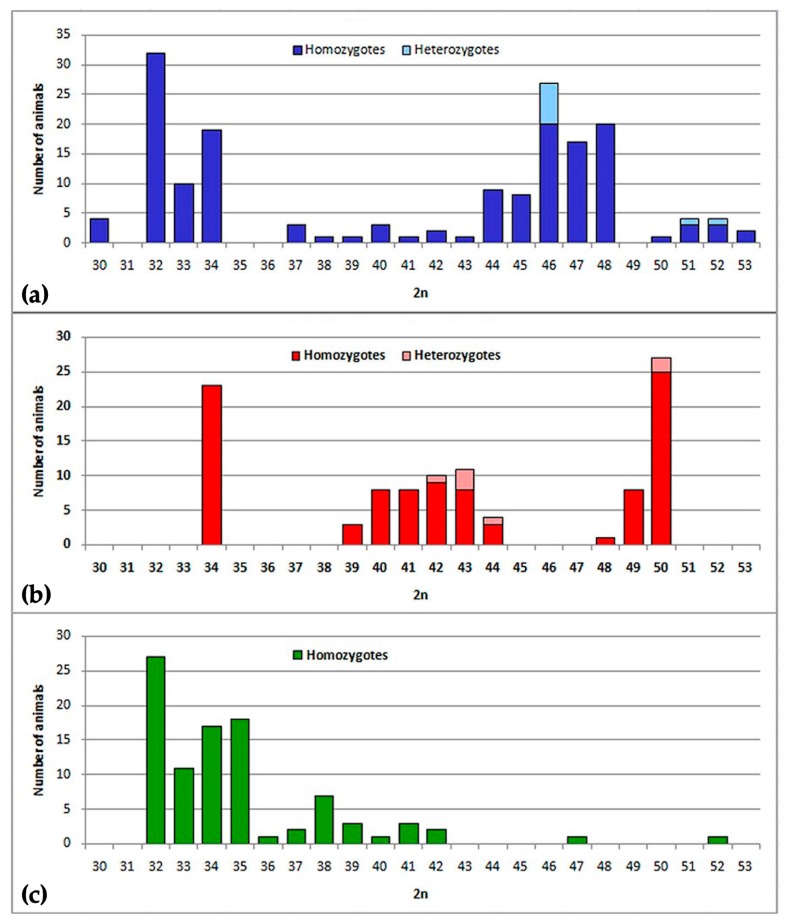
Diploid chromosome number variations in three chromosomal forms: (**a**) chromosome form I, (**b**) chromosome form II, and (**c**) chromosome form III. The abscissa axis indicates the value of diploid numbers of chromosomes, the ordinate axis indicates the number of animals. Saturated colors show the number of homozygotes, light colors show the number of heterozygotes.

**Figure 3 ijms-20-04466-f003:**
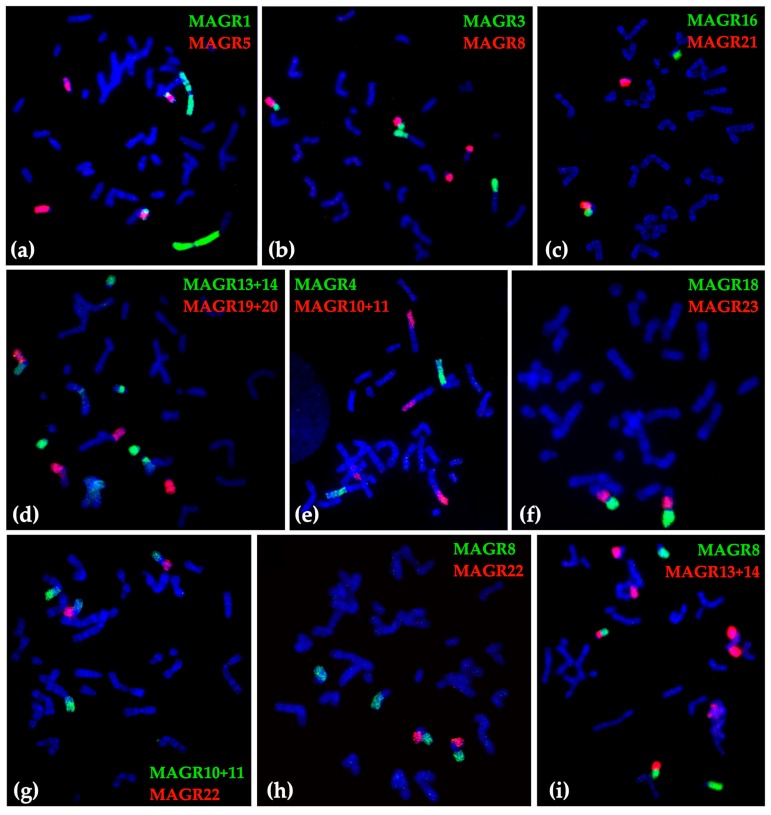
Examples of fluorescent in situ hybridization of *M. agrestis* (MAGR) chromosome-specific probes onto chromosomes of different *E. tancrei* specimens: (**a**) 24914, (**b**) 24920, (**c**) 25621, (**d**) 25620, (**e**) 25617, (**f**) 25621, (**g**) 25621, (**h**) 25614, and (**i**) 24919.

**Figure 4 ijms-20-04466-f004:**
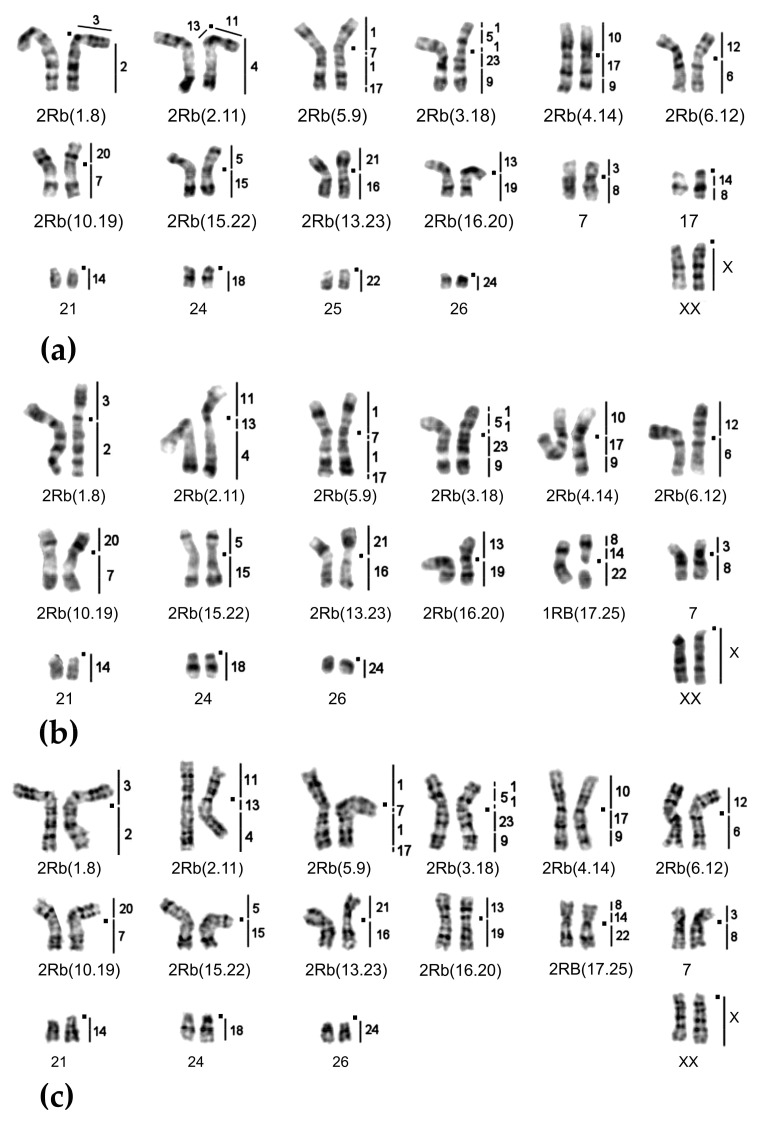
Chromosome form I, G-banded karyotypes: (**a**) 24919♀ 2n = 34, (**b**) 24920♂ 2n = 33, and (**c**) 25613♂ 2n = 32. Black squares mark centromere positions. Vertical bars and the numbers beside them correspond to the localization of *M. agrestis* chromosome segments.

**Figure 5 ijms-20-04466-f005:**
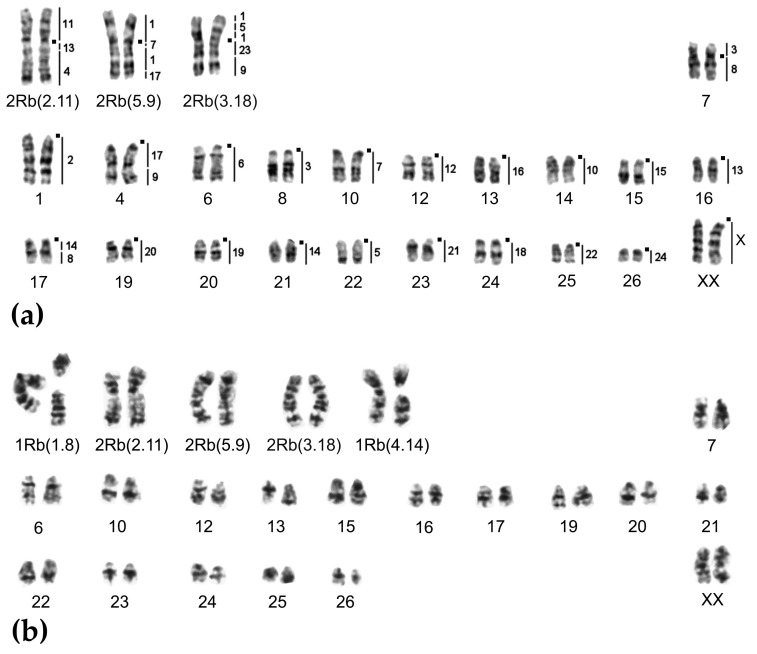
Chromosome form I, G-banded karyotypes: (**a**) 24915♂ 2n = 48, (**b**) 20350♂ 2n = 46. Black squares mark centromere positions. Vertical bars and the numbers beside them correspond to the localization of *M. agrestis* chromosome segments.

**Figure 6 ijms-20-04466-f006:**
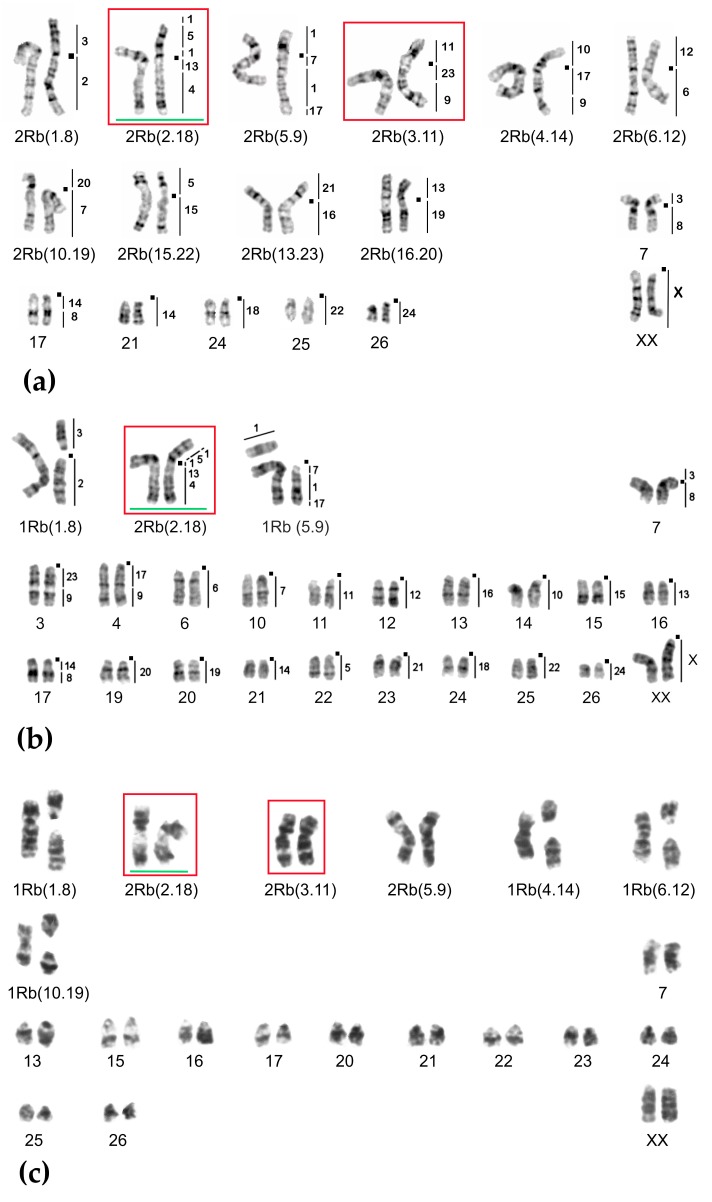
Chromosome form II, G-banded karyotypes: (**a**) 25617♀ 2n = 34, (**b**) 24909♀ 2n = 50, (**c**) 20422♂ 2n = 44. Black squares mark centromere positions. Vertical bars and the numbers beside them correspond to the localization of *M. agrestis* chromosome segments. Rbs specific for chromosome form II are outlined in red. The rearrangements characteristic for chromosome form III are highlighted in green.

**Figure 7 ijms-20-04466-f007:**
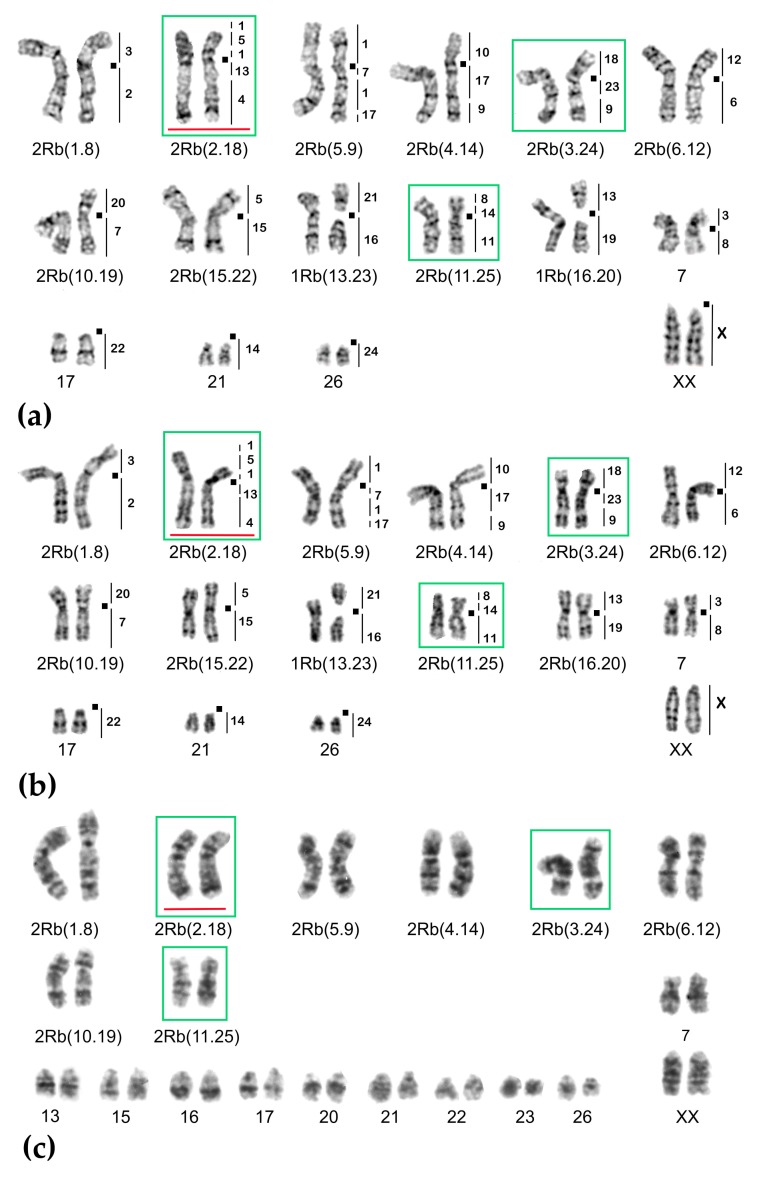
Chromosome form III, G-banded karyotypes: (**a**) 25620♂ 2n = 34, (**b**) 25621♀ 2n = 33, (**c**) 20389♂ 2n = 38. Black squares mark centromere positions. Vertical bars and the numbers beside them correspond to the localization of *M. agrestis* chromosome segments. Robertsonian translocations (Rbs) specific for chromosome form III are outlined in green. The rearrangements characteristic for chromosome form II are highlighted in red.

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
