# Peer review of "Chromosome Translocations as a Driver of Diversification in Mole Voles Ellobius (Rodentia, Mammalia)"

_ijms, 2019, doi:10.3390/ijms20184466_

Round 1

Reviewer 1 Report

In the manuscript “Chromosome translocations as a driver of diversification in mole voles Ellobius (Rodentia, Mammalia)”, authors examined diversification trends within a group of subterranean rodents, the mole voles Ellobius. The previous studies revealed the mechanism of homologous Robertsonian translocations as leading factor of chromosome variability. Here, controversial to previous reports, authors proposed and demonstrated that chromosomal translocations are not always homologous and karyotypes with the same diploid number can carry different combinations of metacentrics. The authors also suggested that the emergence of partial or monobrachial homology might be a strong barrier for hybridization in nature.

In general this is a straightforward manuscript that suggests and proves that chromosomal translocations lead to diversification and speciation of mole voles. Even though the manuscript will attract attention of only few scientists who study the involvement of chromosome changes in the initial steps of specification, it is overall well written.

Minor recommendation:

It might be better for the readers if authors use borders/windows for particularly analyzed G-banded karyotypes, to separate analyzed chromosome sets of a), b) and c) in the Figure 4-7.

Author Response

We tried to follow the suggestion, but the pictures became overloaded with details.

Reviewer 2 Report

The manuscript provides evidence on chromosomal diversification in a rodent Ellobius tancrei in a comparatively small area of the Alai-Pamir. The variation is well documented: sampling was dense, samples sufficient and karyological analyses performed competently. The narrative part needs improvements. English is not always used properly hence the statements are not always easily understandable. The authors frequently do not specify the object of their attention clearly enough, leaving a reader to guess. The aim of the study must be better defined. Chromosomal forms (I, II, III) must be clearly defined and diagnosed. It is confusing the see (Fig.2) that same 2n belong to different Chromosomal forms. Of course, differences are in Rbs, but help a reader and expose this.

Minor remarks

51 proposed modification: referred to as CONSPECIFIC WITH E. talpinus

52-3 in my understanding of the topic, it is not the NF the agent of reproductive isolation, but the position of the centromere (what causes the change in NF). Besides, I would not make the statement so categorical; please consider: which PROBABLY leads to …

60-1 Information in the caption is not sufficient. You use 4 colours but only 3 are explained. Use taxonomic name(s): Map of the distribution of the chromosome forms of Ellobius tancrei found in …. What is the meaning of »E. alaicus« in the map? What is the broken line? Please explain.  Provide, as an inset, the map of wider area showing the position of the study site. It would help a reader if different symbols would be used for the Rbs and for the hybrids. Please, provide also a scale bar. Make sure that all toponyms, mentioned in the text, are shown also in Fig.1.

62 Better avoid passive; who outlined the hypotheses? Name him, her, them: WE OUTLINED …; XY OUTLINED

63 consider insertion: (i) a low-chromosomal form originated FROM THE ANCESTRAL 2N=54 KARYOTYPE by A chain mutation processes

If the hypotheses (lines 62-8) are by the authors, the paragraph 69-71 should go before line 62

76 can you explain the meaning of the numbers

77 »several« proposed to be replaced by »selected«

79 I would delete »significantly«

84 Figure caption should provide more information:

85-6 I do not understand the last sentence of the caption.

98 A reader (like myself) is not familiar with the topography of your study area, hence he/she will not understand why you results is a surprise.

109 “a small amount” – do you mean: few individuals (in low number)?

110-1 the sentence needs to be reworded. Populated the Fergana and Varzob valleys – when? Geographic localities (Fergana, Varzob) are not shown on Fig 1, hence a reader will not easily follow your line of argument.

111 I am not familiar with the term “remote hybridization” – Please explain. Furthermore, do not be enigmatic: ... hybridization with the low-chromosomal form ... Do you mean hybridization between 2n=54 and which 2n???  Please, specify. Next: …. Hybridization …obtained more Rbs in the heterozygous state – obtained from where?

125 »at this point« I don't comprehend the meaning.

128 »We were unable to detect a hybrid zone for E. tancrei and E. alaicus there.« Do you mean: BETWEEN E. tancrei and E. alaicus there.

130-3 “Another chromosome form with 2n = 34” Consider rewording: chromosome form II with 2n = 34 … between the Surkhob and the Obikhingou Rivers, to the west of 70°30'E. The settlements not shown on Fig.1 – plot them, or omit the names from the text.

147 Sangikar River not shown on the Fig.1

150 »described« - consider: characteristic

154 Sangikar and Sorbog Rivers – any of these rivers is shown in Fig.1

156 “low-chromosomal form« - do you mean a chromosomal form with low 2n? »Original« - do you mean the ancestral?

162 “human-made bridges« consider: man-made bridges

182 please, specify which histogram you have in mind

186 »The chromosomal forms« - do you mean: The three …

194-5 »The problem of proving this type of translocations has not yet been solved« - rewording needed.

196 »by this type of translocation« - which type? Please, be specific and name it.

203 »A specific Rb« - please, anme it

203-4 »was fixed in E. alaicus for 30 years« - do you mean: was fixed in E. alaicus for AT LEAST 30 years

216 “the map demonstrates possible geographical connections« - not specified in caption to Fig.1

217-21 I cannot follow the line of argument – something is missing here. How would a local extirpation accelerate evolutionary change?

223-5 I cannot the object the conclusion. But I still do not understand the link between the monobrachial homology and the speciation in E. tancrei. The presence of the monobrachial homology proves little on its own.

Author Response

We have carefully considered the comments while preparing the revised manuscript.

The manuscript provides evidence on chromosomal diversification in a rodent Ellobius tancrei in a comparatively small area of the Alai-Pamir. The variation is well documented: sampling was dense, samples sufficient and karyological analyses performed competently. The narrative part needs improvements. English is not always used properly hence the statements are not always easily understandable. The authors frequently do not specify the object of their attention clearly enough, leaving a reader to guess. The aim of the study must be better defined. Chromosomal forms (I, II, III) must be clearly defined and diagnosed. It is confusing the see (Fig.2) that same 2n belong to different Chromosomal forms. Of course, differences are in Rbs, but help a reader and expose this.

Minor remarks

51 proposed modification: referred to as CONSPECIFIC WITH E. talpinus

Done

52-3 in my understanding of the topic, it is not the NF the agent of reproductive isolation, but the position of the centromere (what causes the change in NF). Besides, I would not make the statement so categorical; please consider: which PROBABLY leads to …

Corrected.

60-1 Information in the caption is not sufficient. You use 4 colours but only 3 are explained. Use taxonomic name(s): Map of the distribution of the chromosome forms of Ellobius tancrei found in …. What is the meaning of »E. alaicus« in the map? What is the broken line? Please explain.  Provide, as an inset, the map of wider area showing the position of the study site. It would help a reader if different symbols would be used for the Rbs and for the hybrids. Please, provide also a scale bar. Make sure that all toponyms, mentioned in the text, are shown also in Fig.1.

The Figure 1 was corrected. We added some toponyms on the map. We also inserted a fragment of the world map to show the position of the study site. It is not possible to point out all the toponyms because the map is too small.

62 Better avoid passive; who outlined the hypotheses? Name him, her, them: WE OUTLINED …; XY OUTLINED

We decided not to change the structure of the sentence.

63 consider insertion: (i) a low-chromosomal form originated FROM THE ANCESTRAL 2N=54 KARYOTYPE by A chain mutation processes

Added.

If the hypotheses (lines 62-8) are by the authors, the paragraph 69-71 should go before line 62

We placed the reference in the sentence before colon.

76 can you explain the meaning of the numbers

 The numbers show the  collection numbers.

77 »several« proposed to be replaced by »selected«

Replaced.

79 I would delete »significantly«

We left this word.

84 Figure caption should provide more information:

The figure caption was expanded.

85-6 I do not understand the last sentence of the caption.

The sentence was deleted.

98 A reader (like myself) is not familiar with the topography of your study area, hence he/she will not understand why you results is a surprise.

The Surkhob River is a rough wide mountain river that a man can't cross except over a bridge.

We corrected as 'Unexpectedly, this form crossed the rough wide mountain river and settled at the southern bank also'

109 “a small amount” – do you mean: few individuals (in low number)?

Corrected.

110-1 the sentence needs to be reworded. Populated the Fergana and Varzob valleys – when? Geographic localities (Fergana, Varzob) are not shown on Fig 1, hence a reader will not easily follow your line of argument.

Corrected as “E. tancrei with 2n = 54 populate large areas to the North, such as the Fergana and Varzob valleys and migration from these regions to the Surkhob River Valley was possible in the past.”

111 I am not familiar with the term “remote hybridization” – Please explain.

The term was introduced by Barbara McClintock.

We deleted 'remote'.

Furthermore, do not be enigmatic: ... hybridization with the low-chromosomal form ... Do you mean hybridization between 2n=54 and which 2n???  Please, specify. Next: …. Hybridization …obtained more Rbs in the heterozygous state – obtained from where?

Corrected.

125 »at this point« I don't comprehend the meaning.

Changed to 'at this site'.

128 »We were unable to detect a hybrid zone for E. tancrei and E. alaicus there.« Do you mean: BETWEEN E. tancrei and E. alaicus there.

Corrected.

130-3 “Another chromosome form with 2n = 34” Consider rewording: chromosome form II with 2n = 34 … between the Surkhob and the Obikhingou Rivers, to the west of 70°30'E. The settlements not shown on Fig.1 – plot them, or omit the names from the text.

We added some of the names and omitted others.

147 Sangikar River not shown on the Fig.1

Corrected.

150 »described« - consider: characteristic

Corrected.

154 Sangikar and Sorbog Rivers – any of these rivers is shown in Fig.1

Corrected.

156 “low-chromosomal form« - do you mean a chromosomal form with low 2n? »Original« - do you mean the ancestral?

The sentence was corrected as: “High heterozygosity supported the presumption of hybridization between the low-chromosomal form III, 2n = 34 with the original form, 2n = 54”

162 “human-made bridges« consider: man-made bridges

Corrected.

182 please, specify which histogram you have in mind

The reference to Figure 2 was added.

186 »The chromosomal forms« - do you mean: The three …

Corrected.

194-5 »The problem of proving this type of translocations has not yet been solved« - rewording needed.

This part was rewritten as: “The problem of explaining the occurrence of these translocations has not yet been solved; nevertheless, the presence of monobrachial homology in some cases is easier to explain by WART.”

196 »by this type of translocation« - which type? Please, be specific and name it.

WART. Corrected.

203 »A specific Rb« - please, anme it

Done.

203-4 »was fixed in E. alaicus for 30 years« - do you mean: was fixed in E. alaicus for AT LEAST 30 years

We repeated the study with an interval of 30 years. So, possibly, the karyotype was changed even faster.

216 “the map demonstrates possible geographical connections« - not specified in caption to Fig.1

The Figure and it’s capture were changed.

217-21 I cannot follow the line of argument – something is missing here. How would a local extirpation accelerate evolutionary change?

Corrected.

223-5 I cannot the object the conclusion. But I still do not understand the link between the monobrachial homology and the speciation in E. tancrei. The presence of the monobrachial homology proves little on its own.

Corrected as “The origin of monobrachial homology is a route to speciation [42]. The emergence of such a homology leads to fertility disorders or sterility of hybrids, which provides isolation of gene pools. Such cases are rare, and E. tancrei appears to be one of the most valuable models for studying chromosomal speciation.”